

# Morphological features and mixing states of soot-containing particles in the marine boundary layer over the Indian and Southern Oceans

Sayako Ueda[1], Kazuo Osada[1], Keiichiro Hara[2], Masanori Yabuki[3], Fuminori Hashihama[4], Jota Kanda[4]

[1] Graduate School of Environmental Studies, Nagoya University, Nagoya, 464-8601, Japan
[2] Department of Earth System Science, Faculty of Science, Fukuoka University, Fukuoka, 814-0180, Japan
[3] Research Institute for Sustainable Humanosphere, Kyoto University, Kyoto, 611-0011, Japan
[4] Department of Ocean Sciences, Tokyo University of Marine Science and Technology, Tokyo, 108-8477, Japan

*Correspondence to*: S. Ueda (ueda-s@isee.nagoya-u.ac.jp) and K. Osada (kosada@nagoya-u.jp)

**Abstract.** Mixing states of soot-containing aerosol particles are important information for the simulation of climatic effects of black carbon aerosols in the atmosphere. To elucidate the mixing states and morphological features of soot-containing particles in remote ocean areas, we conducted onboard observations over the southern Indian Ocean and the Southern Ocean during the TR/V Umitaka-maru UM-08-09 cruise, which started from Benoa, Indonesia on 1 December 2008 via Cape Town, South Africa and which terminated in Fremantle, Australia on 6 February 2009. The light absorption coefficients of size-segregated particles (<0.3 and <1.5 μm diameter) and aerosol number concentrations (0.1–0.5 μm diameter) were measured to assist direct aerosol sampling. Size-segregated aerosol particles were collected for chemical analysis using ion chromatography. For transmission electron microscopy (TEM) analyses with water-dialysis methods, dried submicrometer aerosol particles were collected using a cascade impactor. For detailed individual particle analyses, 13 TEM samples were selected according to their geographical distribution and light absorbing coefficients. Results of water-dialysis analysis show that most particles were water soluble. However, for all TEM samples, rarely particles (2.1% of particles on a TEM sample at a maximum) were found as particles containing insoluble residuals having the characteristic soot shape. For samples collected over the Indian and Southern Oceans at latitudes less than 62°S, some (20–38%) soot-containing particles were found as bare soot. For samples collected near the Antarctic coast (65–68°S, 38–68°E), all soot-containing particles were mixed with water-soluble materials. Furthermore, 56% of soot-containing particles had a satellite structure formed by impaction of droplets such as sulfuric acid. Chemical analysis of submicrometer particles near the Antarctic coast revealed high concentrations of non-sea-salt (nss) $SO_4^{2-}$ and $CH_3SO_3^-$, suggesting that aged soot-containing particles were formed by heterogeneous $SO_4^{2-}$ formation and coagulation of ultra-fine particles over the Southern Ocean.

## 1 Introduction

Soot in the atmospheric aerosol, which is a by-product of fossil fuel (diesel and coal) combustion and open biomass burning, is a carbonaceous material with a deep black appearance to visible solar radiation in the atmosphere (Ramanathan and Carmichael, 2008). Soot has aggregated morphology of globules with diameters of tens of nanometers, consisting of concentrically wrapped graphitic layers (e.g., Pósfai et al., 2004; Murr and Soto, 2005). Carbonaceous materials with a deep black appearance having a large imaginary part of the refractive index are measured optically as black carbon (BC). Although the carbon fractions that are designated by different definitions are far from being the same, those for soot and BC overlap to a great extent (Grencsér, 2004).

In fact, BC in atmospheric aerosols can strongly influence the Earth's radiation budget through atmospheric processes, but also through positive feedback on snow and ice albedo after transport and deposition on the snow surface at mid-to-high latitudes (e.g., Haywood and Boucher, 2000; Hansen and Nazarenco, 2004; Koch and Hansen, 2005; Ramanathan and Carmichael, 2008; Bond et al., 2013). Nevertheless, there are order-of-magnitude disagreements of BC concentrations



between models and observations in remote and upper troposphere air masses (Koch et al., 2009; Schwarz et al., 2010; Schwarz et al., 2013).

Soot particles that are freshly emitted from fossil fuel combustion are attached to or coated with secondary aerosol materials such as sulfates, nitrates, and organics through atmospheric aging processes (Weingartner et al., 1997; Zuberi et al.,

2005). The aging processes alter the particle size, hygroscopicity, and the ability to act as cloud condensation nuclei, eventually reducing the residence time of soot particles in the atmosphere. Some numerical sensitivity studies have pointed out that aging and wet scavenging parameters of soot in the atmosphere are key factors controlling long-range transport and spatial distributions (Koch, 2001; Croft et al., 2005; Stier et al., 2006).

To improve global BC modelling, fundamental information related to aging levels of soot-containing particles is

necessary for the atmosphere in remote or polar regions because of their importance for snow albedo effects caused by BC deposition onto snow or sheet ice. Some reports have described aged soot-containing particles using electron microscopy (Pósfai et al., 1999; Hasegawa et al., 2002; Hara et al., 2003; Vester et al., 2007; Ueda et al., 2011, 2016; Adachi et al., 2014). However, most such observations were limited for sampling locations near source areas. Particularly, knowledge of individual features of soot-containing particles in remote areas of the southern hemisphere remains very poor. Although

some reports of observations have described BC concentrations over marine boundary layers of the southern hemisphere (Moorthy et al., 2005; Evangelista et al., 2007; Sciare et al., 2009) and Antarctica (Wolff et al., 1998; Hansen et al., 2001; Hara et al., 2008; Chaubey et al., 2010; Weller et al., 2013), information related to mixing states of soot has not been shown.

An important reason for the scarcity of data in remote regions is the difficulty of sampling aerosols under clean conditions. In remote areas, the BC concentration is usually quite low. Moreover, the mass proportion is very low compared

to other aerosol components such as sea-salt and sulfates. Therefore, it is difficult to find rare soot particles in many other components, particularly well-aged soot. However, water-dialysis to detect insoluble soot selectively under a microscope is a powerful technique to investigate the mixing states of well-aged soot-containing particles (Okada et al., 1983; Ueda et al., 2011a, 2011b). This method comprises morphological observation and comparison before and after water-dialysis of aerosols. Because soot shows distinctive morphological features and because it is water-insoluble, this method is suitable to

detect soot and to investigate the mixing states with water soluble materials.

For this study, we conducted careful sampling for individual analysis of soot-containing particles onboard the TR/V Umitaka-maru cruises from December 1, 2008 to February 6, 2009 over the southern Indian Ocean and the Southern Ocean. Elucidating the mixing states of soot-containing particles for low-BC concentration areas will be helpful for understanding long-range transport of soot in remote areas, and eventually for understanding the global diffusion processes of BC through

the atmosphere. This study was conducted mainly to ascertain the mixing states of soot-containing particles with water-soluble materials and to assess their relation to morphological features of the mixing materials.

## 2 Field observation and laboratory methods

Atmospheric observations were conducted over the Indian Ocean and the Southern Ocean during the TR/V Umitaka-maru UM-08-09 from December 1, 2008 through February 6, 2009. Figure 1 portrays ship tracks of the TR/V Umitaka-maru

cruise.

### 2.1 Aerosol number–size distribution

A flow diagram of a measurement system and a sampling system for TEM samples is presented in Fig. 2. Number–size distributions of atmospheric aerosol particles were measured using an optical particle counter (OPC, KC-18; Rion Co. Ltd.) similar to the system used for a study described by Ueda et al. (2011a). The OPC measures the number concentrations of



aerosol particles for five size ranges: diameters greater than 0.1, 0.15, 0.2, 0.3, and 0.5 μm. The aerosol measurements were made at relative humidity (RH) of 15–25%, monitored on-line using a data logger (MR6600; Chino Corp.).

## 2.2 Size segregated light absorption coefficient

Light absorption of aerosol particles was measured using two particle soot absorption photometers (PSAP; Radiance Research), as derived from the particle light absorption coefficient at 565 nm wavelength ($b_{abs}$). The values of $b_{abs}$ were corrected based on the method described by Bond et al. (1999). To obtain size information, two impactors (nozzle diameters 0.4 and 1.2 mm, flow rates 0.8 and 1.0 L min$^{-1}$, respectively) and two two-way valves were used for a PSAP to separate particles with diameter larger than 0.5 μm and larger than 1 μm. The other was used to measure light absorption for total particles. The valves were changed every 6 min with integration time of 1 min for one photometric value. It is noteworthy

that the values of maritime $b_{abs}$, especially those of the larger size range, can be overestimated by effects of scattering by sea salts and other components on the filter. However, correction of the scattering effect was difficult because $b_{abs}$ over remote ocean region was usually quite low. For this study, values of $b_{abs}$ less than 1 μm were used mainly for evaluating the relative variation with attention to overestimation, and as an index of clean sampling of TEM samples aboard the ship.

## 2.3 Data screening of aerosol number concentration and light absorption

Condensation nuclei (CN) concentrations were measured using a CN counter (3781; TSI) for particles with diameter greater than 10 nm (Fig. 2). A two-way valve and a diffusion screen to cut particles smaller than 20 nm were used for the CN counter to obtain information about the nucleation particle size. The valves were changed every 10 min. To discuss background aerosol particles specifically, we carefully removed data showing contamination by ship exhaust using CN concentration. As an example, CN concentrations before and after data screening are presented in Fig. 3. CN concentrations

before screening sometimes increased dramatically for a short time because of local contamination. We applied restrictions using CN data to the aerosol number size distribution and $b_{abs}$, and eliminated suspicious data that did not conform to the following: (1) standard deviation of CN concentrations during 10 min is less than 10% of the median value and (2) CN concentration during 1 min is lower than 1.2 times the median value for 10 min, in accordance with Ueda et al. (2016). As a result of screening according to (1) and (2), periods of rapid CN increase and high CN concentration were omitted (Fig. 3).

For the data of OPC and PSAP, the periods of (1) and (2) were not used. Consequently, the aerosol concentration and $b_{abs}$ data showed no sudden change, as described later in section 3.1.

## 2.4 Ionic constituents of size-segregated aerosol particles

Size-segregated aerosol samples were collected using a three-stage impactor with a back-up filter for 24 h intervals at a flow rate of 20 L min$^{-1}$. The air sampler was placed in a weather shield at the middle of the front end on the uppermost deck. Air

sampling was controlled as relative wind speed (>2 m/s) and direction (from bow) to avoid contamination from the ship's exhaust. To prevent contamination from the ship's boundary layer, the inlet of the weather shield was designed to protrude toward the bow from the edge of the uppermost deck. A similar arrangement and strategy of aerosol sampling was used by Kawakami et al. (2008). The estimated 50% cut-off diameter for Stage 1 was 8 μm, with 2 μm for Stage 2, and 0.2 μm for Stage 3 at a flow rate of 20 L min$^{-1}$. In this study, the Stage 3 results were used for comparison with other aerosol data. The

substrate of the first impactor stage was a PTFE filter (47 mm diameter; Advantec Toyo Kaisha Ltd.) with a 15 mm diameter hole at the center. Nuclepore filters (25 mm diameter, 110606; Whatman plc.) were used as sampling substrates for the second and third stages of the impactor. The back-up filter was a 47-mm-diameter PTFE filter (nominal pore size of 1.0 μm; Advantec Toyo Kaisha Ltd.). These samples were kept in a freezer until laboratory analyses. Filter samples were analyzed using ion chromatography (DX-120; Dionex Corp.) after extraction using 14 mL of ultrapure water (18.2 MΩ). Analytical

conditions and procedures were described by Hara et al. (2004).





### 2.5 Samples of morphological particle analyses using an electron microscope

Aerosol particles were collected for morphological particle analysis using a transmission electron microscope (TEM). To analyze morphological features of aerosol particles, dried (RH <20%) aerosols were collected using cascade impactors (50% cut-off diameters of the three stages 1, 2, and 3 were, respectively, 0.5 μm, 0.3 μm, and 0.2 μm) on carbon-coated
nitrocellulose (collodion) films. Aerosol samples were collected for 20–70 min at a flow rate of about 0.6 L min$^{-1}$. In this study, samples of stages 1 and 2 were used for analyses. To control the suitable surface density of particles on sampling substrate for observation by TEM analysis, the sampling time was controlled according to aerosol number concentrations. TEM samples were taken at about 2–5 samples per day based on aerosol concentrations and light absorption of aerosol particles. For analysis of atmospheric aerosols, locally contaminated aerosol samples were eliminated by reference to the CN
concentration during TEM sample collection from the same inlet tube. The TEM samples were stored under dry conditions at room temperature until TEM analyses were conducted at Nagoya University.

To obtain clear shape and height information of individual particles on the collection surface, particles collected on the carbon film were coated with a Pt/Pd alloy at a shadowing angle of 26.6° (arctan 0.5). The Pt/Pd coating thickness was about 7 Å. The particles were photographed using TEM (JEM-2010; JEOL) at 2000× magnification. The collection film is
15 regarded as a semipermeable membrane. Therefore, a water dialysis technique (Mossop, 1963; Okada, 1983; Okada and Hitzenberger, 2001; Ueda et al., 2011ab) was applied to aerosol samples to remove water-soluble materials from individual aerosol particles through the film after they were photographed. The electron microscopic grid with particle samples was floated on ultrapure water at room temperature (about 25°C) for 3 hr with the collection side facing upward. The water-insoluble residues after dialysis were coated again perpendicularly to the previous coat of a Pt/Pd alloy to differentiate the
20 particle height and two-dimensional morphology after water dialysis. They were then rephotographed using TEM.

The negative films were scanned and recorded with resolution of 1200 dpi. The scanned image was processed using image analysis software (Win Roof; Mitani Corp.) to estimate the projected area of particles ($S$). The volume of each particle before and after water dialysis was calculated from measurements of S. Also, the projected area diameter of the particles was estimated from $S$. The mixing states of individual particles with respect to water solubility were obtained by comparing
electron micrographs of the same field of the collecting surface taken before and after water dialysis. For morphological analysis using water dialysis, 365–6270 particles per sample were compared before and after water dialysis.

### 2.6 Air mass backward trajectories

Air mass backward trajectories were analyzed to investigate their relation to observed size distributions and mixing states of aerosol particles. The backward trajectory and precipitation data were computed using the Hybrid Single-Particle Lagrangian
Integrated Trajectory (HYSPLIT) model developed by the National Oceanic and Atmospheric Administration (NOAA) Air Resources Laboratory (ARL) (Stein et al., 2015; Rolph et al., 2017). The setting of the trajectory duration, starting height, vertical mode calculation method, and dataset were chosen respectively as 10 days, 500 m above sea level, model vertical velocity, and GDAS meteorological Data.

### 3 Results and Discussion

#### 3.1 Temporal variation of aerosol parameters

Figures 4 and 5 portray temporal variations of the size-segregated volume concentration (0.1−0.5 μm diameter) of aerosol particles (Figs. 4a and 5a), the $b_{abs}$ of the size-segregated aerosol particles ($D$: <0.5, 0.5–1.0, >1.0 μm) (Figs. 4b and 5b), the ratio of the $b_{abs}$ of $D$<0.5 μm to $D$<1.0 μm aerosols (Figs. 4c and 5c) and wind speed and direction (Figs. 4d and 5d),
respectively, for Benoa – Cape Town and Cape Town – Fremantle. The volume concentrations were calculated from the



number concentrations for the respective size ranges of the OPC and their geometric mean diameters, assuming spherical particles. Contaminated data were removed from dataset based on the screening method. The fog, rain, and snow periods are shown at the top of Figs. 4a and 5a.

The total volume concentrations of aerosols over remote ocean areas were usually 100–500 µm³ L⁻¹. They were high,

about 1000 µm³ L⁻¹, near South Africa (28–29 December 2008, 1–2 and 7–8 January 2009). Throughout the observation period, aerosol volume concentrations of 0.15–0.3 µm diameter comprised about half of the total volume concentration.

The highest $b_{abs}$ (>1.0×10⁻⁶ m⁻¹ at $D$ <1 µm particle) was observed near the coast of South Africa (7 January 2009), but $b_{abs}$ values soon decreased with distance from the coast. The ratio of 0.5 µm to 1.0 µm particles also tended to decrease concomitantly with increasing distance from the coast (7–8 January 2009). High values of $b_{abs}$ (>0.4×10⁻⁶ m⁻¹ at $D$<1 µm

particle) were observed on 13, 20–25 December 2008, 10–11, 13–16 and 28–31 January 2009, and 1 February 2009. Near the Antarctic coast on 16–28 January 2009, $b_{abs}$ was mostly approx. 0.1×10⁻⁶ m⁻¹ (0.3×10⁻⁶ m⁻¹ at most) for $D$<1 µm particles. The ratio of 0.5 µm to 1.0 µm was high. Some observation studies have measured BC concentration over oceans of the southern hemisphere (Moorthy et al., 2005; Evangelista et al., 2007; Sciare et al., 2009) based on light absorption measurements. Moorthy et al. (2005) measured a BC value over the Arabian Sea, the tropical Indian Ocean and the Southern

Ocean. The values of BC remained <50 ng m⁻³ and remarkably steady (in space and time) in the Southern Ocean (20–56°S, 42–60°E) during January–March. Evangelista et al. (2007) measured BC values at the southern East Atlantic coast at latitude 22°–62°S. The values of BC at 50°S–62°S were less than 40 ng m⁻³. Sciare et al. (2009) conducted long-term observations of filter-based monitoring of carbonaceous aerosols at Amsterdam Island (37°S, 77°E). They reported that BC concentrations were among the lowest reported for a marine atmosphere, with monthly mean levels ranging from 2–5 ng C/m³ during

summer. Based on light absorption, BC concentrations in Antarctica have been reported from some observation studies (Wolff et al., 1998; Hansen et al., 2001; Hara et al., 2008; Chaubey et al. 2010; Weller et al., 2013). According to those reports, BC concentrations for summer have been reported as 0.5–5.0 ng m⁻³ at Halley Station (75°S, 26°E) by Wolff et al. (1998), 20–300 ng m⁻³ at McMurdo Station (78°S, 167°E) by Hansen et al. (2001), less than 10 ng m⁻³ at Syowa Station (69°S, 39°E) by Hara et al. (2008), 4–19 ng m⁻³ at Larsemann Hills (69°S, 77°E) and 20–157 ng m⁻³ at Maritri (70°S, 12°E)

by Chaubey et al. (2010), and approx. 2 ng m⁻³ monthly median at Neumayer Station (70°S, 8°W) by Weller et al. (2013). For Maitri, Chaubey et al. (2010) pointed out that the considerable impact of local pollution by human activities must be considered along with results. If a mass-specific absorption cross section of BC were assumed as 10 m² g⁻¹ (Hansen et al., 1984; Gelencsér, 2004), then the BC mass concentration in this study was calculated roughly as >100 ng m⁻³ near the coast of South Africa, but usually 20–60 ng m⁻³ in the remote ocean, showing a similar BC level to those of reports by Moorthy et

al. (2005) and by Evangelista et al. (2007). In addition, the calculated BC concentration near Antarctica (65–68°S, 38–68°E) was about 10 ng m⁻³ (30 ng m⁻³ at most), showing a low level similar to data from other reports of coastal areas of Antarctica (McMurdo Station, Syowa Station, Larsemann Hills and Maritri).

Wind speeds (Fig. 4d) varied during observation periods: 0–23 m s⁻¹. High wind speeds (>15 m s⁻¹) were often observed at latitudes of 40–60°S (21–23 December 2008, 11–13 January 2009, and 1 February 2009). Wind speeds over the Southern

Ocean near the coast of Antarctica (16–27 January 2009) were mostly lower than 10 m s⁻¹, and were often less than 5 m s⁻¹. The mass concentrations of sea-salt aerosol particles derived from the ocean are known to correlate well with wind speed (Lewis and Schwartz, 2004). Although the aerosol concentrations of $D$>0.3 µm were often higher under high wind speed conditions (e.g. 22 December 2008), no correlation was found between wind speed and aerosol concentration.

**3.2 Relation between backward air mass trajectory and horizontal distribution of absorption coefficient and nss-K⁺**

Figure 6 portrays 10-day backward air mass trajectories. Calculations started from 500 m above sea level over the location at noon, local time every day. According to starting areas, trajectory analyses were grouped into six zones (Figs. 6a to 6f). The air masses of the eastern Indian Ocean (2–18 December 2008) were derived from south subtropical or more southern areas,





moving counterclockwise to the observation sites (Fig. 6a). Air masses of the western Indian Ocean (19–30 December 2008, Fig. 6b), south from South Africa of 50–65°S (10–15 January 2009, Fig. 6d), and southwest from Australia (27–31 January and 1, 2 February 2009, Fig. 6f) were derived from southwest of the Atlantic Ocean. Some of them passed around the area south of South America. The air masses from near the coast of South Africa (31 December 2008, 1, 8 and 9 January 2009, Fig. 6c) passed the coast of South Africa, moving counterclockwise from the south. Air masses from the Southern Ocean near Antarctica (16–26 January 2009, Fig. 6e) originated from the Antarctic coast.

Figure 7a portrays a horizontal distribution of $b_{abs}$ for particles with less than 1.0 μm diameter. The $b_{abs}$ around South Africa was higher. The high values ($>4\times10^{-7}$ m$^{-1}$) were often observed in the western Indian Ocean (35–45°S 35–45°E on 20–25 December 2008), south of South Africa (50–65°S 26–38°E on 10–11, 13–16 January 2009), and southwest from Australia (45–63°S 80–98°E on 28–31 January and 1 February 2009). For these periods, backward air trajectories were mostly derived from south of South America. In the subtropical area of South America, biomass burning was often observed by satellite (Edward et al., 2006; Giglio et al., 2006; Chen et al., 2013). Figures 7b and 7c respectively portray horizontal distributions of non-sea-salt potassium (nss-K$^+$) and oxalate concentrations in aerosol particles with sizes of 0.2–2 μm. These species are regarded as originating mainly from biomass burning (Andreae, 1983; Kawamura and Kaplan, 1987, Narukawa et al., 1999). Although the nss-K$^+$ concentrations were below the detection limit (black circle in Fig. 7b) in broad remote ocean areas, discernible nss-K$^+$ ($>3$ ng m$^{-3}$) concentrations were observed on 20–23 and 25 December 2008 (western Indian Ocean), on 9, 12–16 January 2009 (Southern Ocean), and on 31 January 2009 and 1 February 2009 (region southwest of Australia). High concentrations ($>5$ ng m$^{-3}$) of oxalate were also observed in the Indian Ocean on 13–14, 19–20, and 22–23 December 2008. Dates of higher concentrations of nss-K$^+$ or oxalate mostly coincided with higher $b_{abs}$. Some reports of observation have described plumes from South America and southern Africa at Syowa Station (Hara et al., 2010) and Troll Research Station (Fiebig et al., 2009) in Antarctica during winter–spring, when biomass burning is active in the southern hemisphere. However, the season in this study was summer. Most 10-day air mass trajectories did not come directly from subtropical continental areas of South America. In this study, identification of the source was unfortunately difficult by trajectory, but the result obtained for nss-K$^+$ and oxalate suggests that the air mass was partially influenced by biomass burning.

### 3.3 TEM analysis

### 3.3.1 Samples

According to the geographical area of sample collection and $b_{abs}$, 13 samples were analyzed for TEM analysis. The starting times of sample collection are indicated by arrows A–D in Fig. 4a and by arrows E–M in Fig. 5a. Sample details are presented in Table 1. Sampling locations and 10-day backward trajectories of air parcels for samples A–M are portrayed as Fig. 8. Trajectories were started at 500 m above sea level at the sampling site. Samples A–D and G–J were collected under conditions with high $b_{abs}$ over remote areas of the Indian Ocean and the Southern Ocean. Samples E and F were collected near South Africa. Samples K–M were collected over the Southern Ocean near the Antarctic coast.

Samples were classified into five groups referring to the area. Sample A was collected over the eastern Indian Ocean (classified as group 1). Only one sample was adequate for TEM analysis because of frequent contamination in the eastern Indian Ocean caused by surrounding ship activities. Samples B, C, and D were collected over the western Indian Ocean and were classified as group 2. Samples E and F were collected near southern Africa (classified as group 3). Samples G, H, and I were collected at 50–62°S south of South Africa (classified as group 4). Samples J, K, L, and M were collected over the Southern Ocean near the Antarctic coast (classified as group 5).



### 3.3.2 Morphological features and mixing states

Figure 9 shows TEM images of samples A, C, E, H, J, and M, and an example (lower part) of analysis at the same magnification using water dialysis before and after treatment of sample J. Shapes of particles in these samples were round, dome-like, or rotundate rectangular (examples indicated by blue arrows in Fig. 9). According to earlier studies based on elemental analysis using an energy-dispersive X-ray spectrometer, such particles were often recognized as sulfate-rich particles (Li et al., 2003; Li and Shao, 2010; Ueda et al., 2011b, 2014). Particularly, rotundate rectangular particles were identified as ammonium sulfate particles based on selected-area electron diffraction analysis (Ueda et al., 2011b). Some crystalline coarse particles were also found in group 1–4 samples (samples A–I) (examples are indicated by red arrows in Fig. 9). These particles showed a stronger contrast with the collection film and had larger diameters than those of sulfate-like particles. Most of them had cuboidal shape, which is morphological feature of sea-salt particles. By contrast, such sea-salt-like particles were rarely found in samples (samples J–M) of group 5 collected under lower wind speed conditions near the coast of Antarctica. In samples of K, L, and M, most particles had a satellite structure, resembling sample M of Fig. 9. Such particles having a satellite structure were also found in the samples H–J collected over the Southern Ocean, as indicated by green arrows. The satellite structure is typically formed by impaction of sulfuric acid ($H_2SO_4$) droplets (Waller et al., 1963; Frank and Lodge, 1967; Gras and Ayers, 1979). Some reports have indicated that the satellite ring shape was correlated with the degree of ammonization: a single ring for bisulfate ($NH_4HSO_4$) and multiple rings for pure sulfuric acid (Bigg, 1980; Ferek et al., 1983; Ueda et al., 2011b). Although some rotundate rectangular particles having a satellite structure were also found, such particles would be acidic droplet particles neutralized by the addition of ammonium after collection. In samples K, L, and M, which were collected closer to the coast of Antarctica, most particles showed a satellite structure with multiple rings, although the satellite particles were fewer in samples H, I, and J than in samples K–M.

Water-dialysis analysis reveals that most aerosol particles (i.e., rounded, dome-like rectangular and crystalline-coarse particles, and the satellite particles) were almost water soluble. However, quite rarely, insoluble residuals remained after water dialysis. Some of them revealed aggregations of globules with diameter less than 50 nm, which is characteristic of soot particles (Janzen, 1980; Pósfai et al., 2004; Murr and Soto, 2005). In this study, particles containing such water-insoluble aggregated residuals were regarded as soot-containing particles. Numbers of soot-containing particles were 1–10 per sample (Table 1). The number fractions of soot-containing particles to total particles were 1% or less. However, that in sample H was 2%. Although the trajectory did not pass over subtropical areas of South America (Fig. 8), the sampling site of sample H coincided with higher nss-$K^+$ concentrations and higher $b_{abs}$, suggesting some influence from biomass burning.

### 3.3.3 Features of soot-containing particles

Figure 10 presents representative examples of electron micrographs of soot-containing particles before and after water dialysis at the same magnification. In sample D collected over the western Indian Ocean, soot-containing particles of two types were found: bare type soot consisting of water-insoluble soot only (i.e., externally mixing of soot) and internal mixtures of soot and water-soluble materials. Soot-containing particles of both types were also found in sample H. Some particles internally mixed with soot had the satellite structure in sample H, suggesting the existence of acidic droplets. In sample M, all soot was internally mixed with the satellite particles.

Table 2 presents the number and a percentage of each type of soot-containing particles and average values of parameters of soot-containing particles (diameters of soot-containing original particles and inner soot core, and diameter ratios of soot to original particle) for each group. The data are total figures for stages 1 and 2. Figure 11 portrays scatter plots of particle sizes (projected area diameters) before (soot-containing particle: $D_{particle}$) and after (soot: $D_{soot}$) water dialysis of soot-containing particles for groups 1–5. Most $D_{soot}$ particles were of 0.1–0.5 μm diameter (Fig. 11). Although some soot-containing particles had super-micrometer size, $D_{particle}$ were mostly submicrometer size.





In samples of groups 1–4, 20–38% of soot-containing particles were found as bare soot (Table 2). More than half of soot-containing particles of groups 1–4 had diameter $D_{particle}$ less than twice of $D_{soot}$ (Fig. 11). Averaged values of the ratio of $D_{soot}$ to $D_{particle}$ were 0.41–0.78 (Table 2). Averaged values of $D_{particle}$ were largest for group 5 (Table 2). This result disagreed with the high $b_{abs}$ ratios of 0.5 μm to 1.0 μm near Antarctica, as explained in section 3.1. Comparison between results of

PSAP and morphological data presented some difficulties. The number and volume of soot were quite low in remote ocean areas, especially in the Southern Ocean, compared to the other aerosol materials. Therefore, the possibility that $b_{abs}$ of each size range was overestimated by scattering particles was undeniable. In addition, accurate estimation of the diameters of the satellite and liquid particles was difficult. For that reason, the size would differ from the aerodynamic size used in measurement of $b_{abs}$. This section presents discussion of the morphological features of soot-containing particles.

Soot particles freshly emitted by fossil fuel combustion were found as bare type (Weingartner et al., 1997). However, aged soot particles often showed coating of large amounts of secondarily formed materials (Pósfai et al., 1999; Hasegawa et al., 2002; Ueda et al., 2011, 2016; Adachi et al., 2014). In this study, the air masses for groups 1, 2, and 4 remained over remote ocean areas without passing continental areas (Fig. 8). Nevertheless, a certain number of bare soot particles and less-coated soot particles were found in these samples. Such bare soot might have originated from distant ships (e.g. cargo ships)

in the same region. However, locations of our TEM sample collection were far from a major ship route located from north of the Malacca Strait to south of Madagascar, except for sample A (Tournadre, 2014). For groups 2 and 4, although the backward trajectory passed a major ship route a few days before for sample B and 10 days before for sample G, that for the other samples did not. Therefore, contributions from distant ships cruising along the major traffic route on days near the sampling days are regarded as quite low, except for group 1 and sample B.

Internally mixed soot particles with soluble materials can be scavenged preferentially from the atmosphere by cloud and rain processes. This mechanism can also lead to a higher likelihood of smaller and hydrophobic soot particles surviving longer in the atmosphere. Studies of partitioning of soot particles between cloud droplets and cloud interstitial particles at high altitudes have demonstrated that soot particles remained in interstitial particles and that sulfate particles were more likely to be scavenged to cloud droplets (Hallberg et al. 1992, 1994; Kasper-Giebl et al., 2000; Hitzenberger et al., 2001).

Additionally, individual particle analysis using water dialysis showed that bare soot particles are abundant in cloud interstitial particles under high precipitation (2–6 mm hr$^{-1}$) at a high elevation mountain site (Ueda et al. 2011a). Precursor gases for aging particles can also be scavenged by wet processes. Therefore, aging of soot particles will occur slowly in a clean air mass over remote ocean areas. Particularly, weather around 50°S was often rainy and stormy during the cruise. During this study, fog and rain events were observed frequently near the regions of sampling sites G2 (Fig. 4) and G4 (Fig.

5). Therefore, bare soot particles found in these samples might have come through these wet scavenging processes.

All soot-containing particles in samples of group 5 collected near the Antarctic coast were mixed internally with water-soluble materials. Although $D_s$ were less than 0.5 μm, most soot-containing particles had $D_{soot}$ greater than twice the value of $D_{soot}$ (Fig. 11). Average ratios of $D_{soot}$ to $D_{particle}$ were 0.31 (Table 2). The abundance of soot-containing particles with the satellite structure was 56% in group 5. Such soot-containing particles having a satellite structure were found only in samples

obtained at 60°S and 68°S. Figures 11a and 11b respectively portray horizontal distributions of $CH_3SO_3^-$ and $nss\text{-}SO_4^{2-}$ concentrations for 0.2–2 μm particles. Although $nss\text{-}SO_4^{2-}$ has both natural and anthropogenic sources, $CH_3SO_3^-$ in marine aerosols is derived from the oxidation of dimethyl sulfide (DMS) emitted by marine phytoplankton (Savoie et al., 1992). Actually, the $CH_3SO_3^-$ concentrations in Fig. 12 were higher south of 50°S and were quite low at lower (<30°S) latitudes. Sporadic high concentrations of $CH_3SO_3^-$ (>150 ng m$^{-3}$) were found in coastal areas of Antarctica. High $nss\text{-}SO_4^{2-}$

concentrations over the Southern Ocean correspond well with the high concentration for $CH_3SO_3^-$. Figure 13 presents a scatter plot of concentrations of $CH_3SO_3^-$ and $nss\text{-}SO_4^{2-}$. For the Southern Ocean (area larger than 60°S), $nss\text{-}SO_4^{2-}$ concentration showed good correlation with $CH_3SO_3^-$. In addition, the number size distribution of sub-micrometer aerosols over the Southern Ocean was characterized by abundant smaller particles (Fig. 5a). These results suggest strongly that sub-





micrometer aerosols over the Southern Ocean are dominated by secondary particles produced by DMS oxidation after local marine biogenic emission. Because DMS production and oxidation are both high in summer in the Southern Ocean, condensation of oxidation products by gaseous DMS and coagulation of the ultra-fine particles engender transformation of soot particles and the forming of soot-containing particles coated by sulfuric acid or methanesulfonic acid droplets.

**4 Conclusions**

To elucidate the mixing states and morphological features of soot-containing particles in remote marine areas, we conducted ship-borne aerosol observations over the Southern and southern Indian oceans during the 27th Umitaka-maru cruise. After TEM samples for individual particle analysis using water-dialysis were obtained, 13 samples were chosen for detailed analysis by sampling location and $b_{abs}$.

Water-dialysis examination revealed that many particles on the 13 TEM samples contained water soluble materials. As the particle number fraction, 0.03–2.11% of particles on the samples contained chain-like insoluble residuals (soot) of 0.1– 0.5 μm diameter.

For samples collected over the southern Indian Ocean and northern Southern Ocean, north of 62°S latitude, 20–38% of soot-containing particles were found as bare soot. Preferential scavenging of aged soot-containing particles might be a

15 mechanism supporting the existence of bare soot over remote ocean areas. The bare soot particle origin remains unclear. The ratio of diameter of soot / soot-containing particles for samples obtained near the Antarctic coast (0.31) was smaller than those of the other samples (0.41–0.78), suggesting thicker coating than in other places.

This study used shipboard observation and microscopic analysis to investigate individual features of soot-containing particles in a maritime atmosphere. Results indicate that bare soot particles were present over the remote Indian Ocean and

20 northern parts of the Southern Ocean, and that aged soot-containing particles were transformed by soluble materials derived from DMS oxidation over the Southern Ocean. Differences of mixing states and morphological features on soot-containing particles for various ocean areas must be considered for evaluation of long-range transport of soot particles and for simulating proper climatic effects of soot-containing particles in the atmosphere.

**Acknowledgments**

We are indebted to staff members of the Umitaka Maru or assisting our work on board. We wish to express gratitude to Mr. T. Goto at the Technical Center of Nagoya University for technical advice on electron microscopy. We gratefully acknowledge the NOAA Air Resources Laboratory (ARL) for providing the HYSPLIT transport model (http://www.arl.noaa.gov/ready.html). This work was performed with the support of a Grant-in-Aid for Environmental

Research provided by the Steel Foundation for Environmental Protection Technology, and the support of a Grant-in-Aid for Scientific Research in Priority Areas, Grant No. 18067005 (W-PASS), provided by the Ministry of Education, Culture, Sports, Science and Technology, Japan, and by Grants-in-Aid for Scientific Research (B) 20310009 and 15H02803, and (A) 20244078 from the Ministry of Education, Culture, Sports, Science and Technology. This research is a contribution of IGBP/SOLAS activity.

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




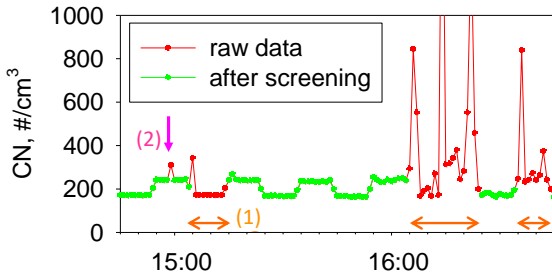

5   **Figure 3: Results for CN concentrations measured before and after screening. Data for periods of orange arrows show removal by screening method (1). Data indicated by the pink arrow are data removed by screening method (2).**



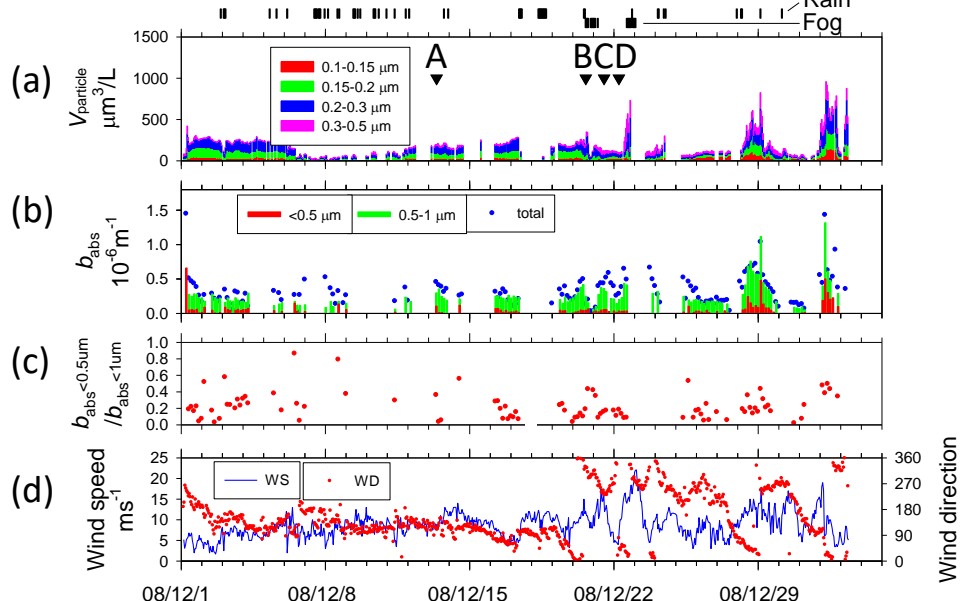

**Figure 4: Temporal variations of (a) volume concentration of aerosol particles and start times of TEM sampling (A–M with arrows), (b) absorption coefficient $b_{abs}$ of size-segregated aerosols, (c) ratio of $b_{abs}$ of <0.5 µm to $b_{abs}$ of <1.0 µm particles and (d) wind speed and direction from Benoa to Cape Town. Vertical marks above (a) show periods of rain and fog.**





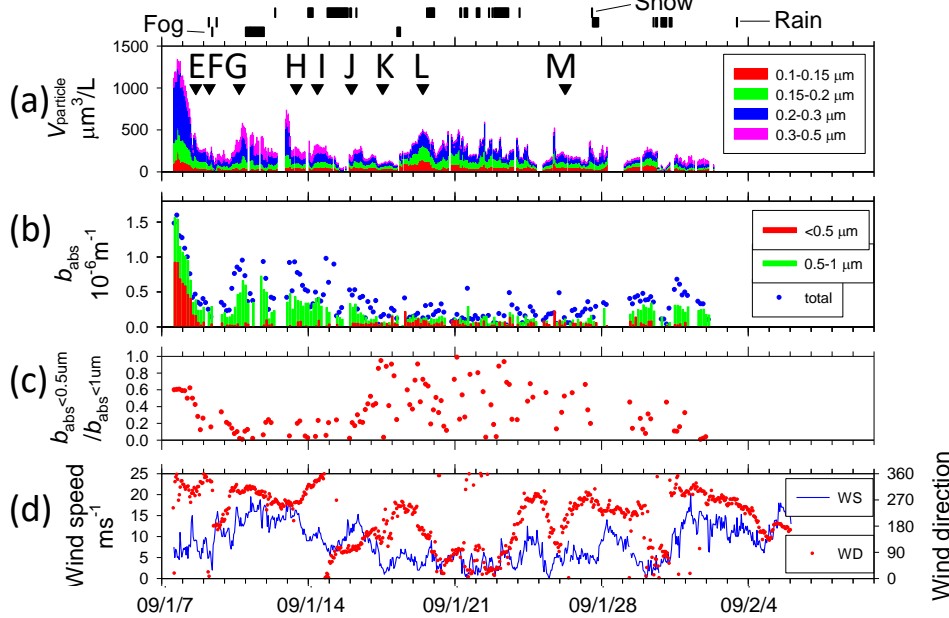

**Figure 5: Temporal variations of (a) volume concentration of aerosol particles and start times of TEM sampling (A–M with arrows), (b) absorption coefficient *b*<sub>abs</sub> of size-segregated aerosols, (c) ratio of *b*<sub>abs</sub> of <0.5 µm to *b*<sub>abs</sub> of <1.0 µm particles, and (d) wind speed and direction from Cape Town to Fremantle. Vertical marks above (a) show periods of snow, rain, and fog.**





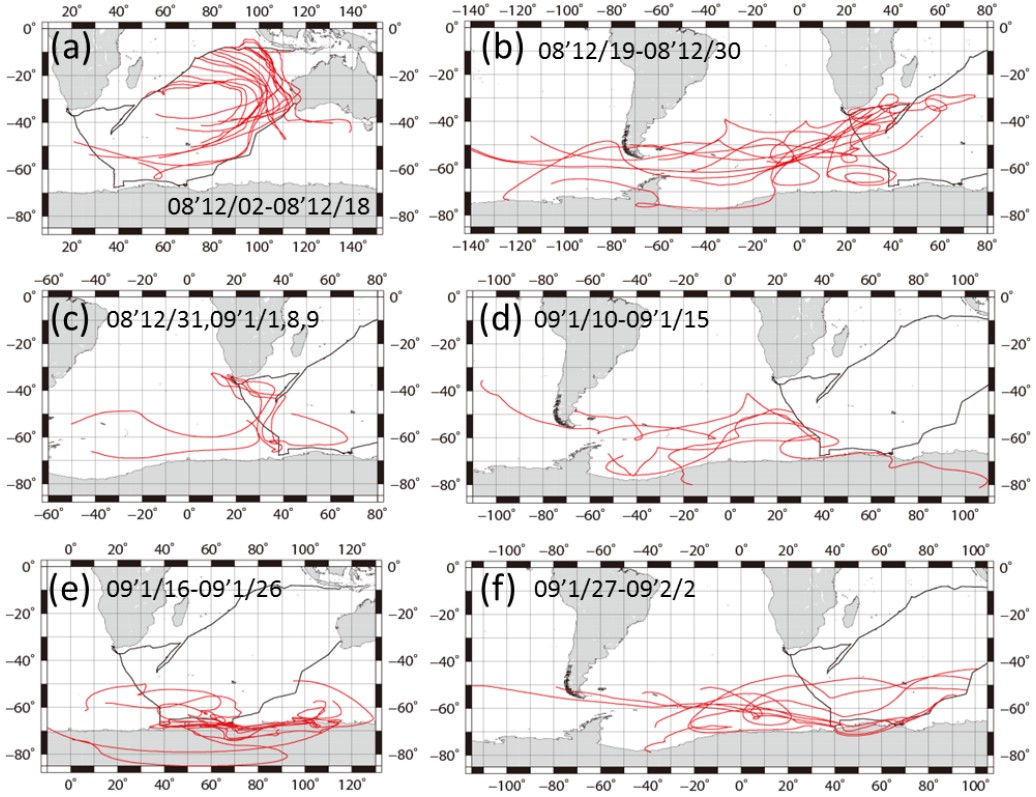

5  **Figure 6: 10-day backward air mass trajectory along the ship tracks. Calculations started from 500 m a.s.l. above the site every day at noon, local time. Red lines show backward air mass trajectories. Black lines show ship tracks.**





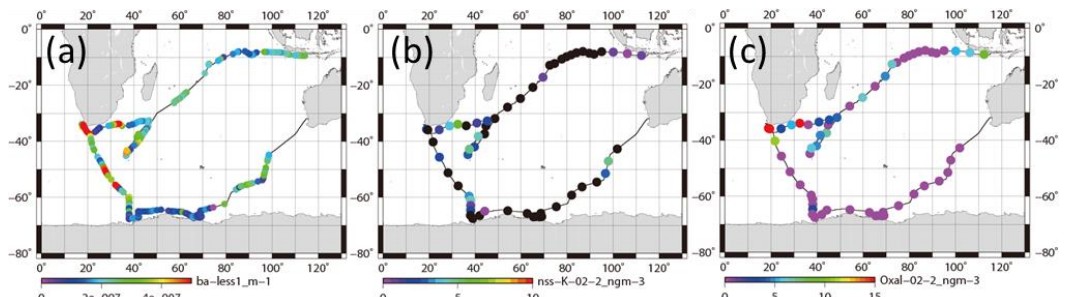

5    **Figure 7: Horizontal variation of (a) absorption coefficient of particles smaller than 1 μm, (b) mass concentration of nss-K⁺ for 0.2–2 μm aerosols, and (c) mass concentration of oxalate for 0.2–2 μm aerosols. Black circles show values below the detection limit.**



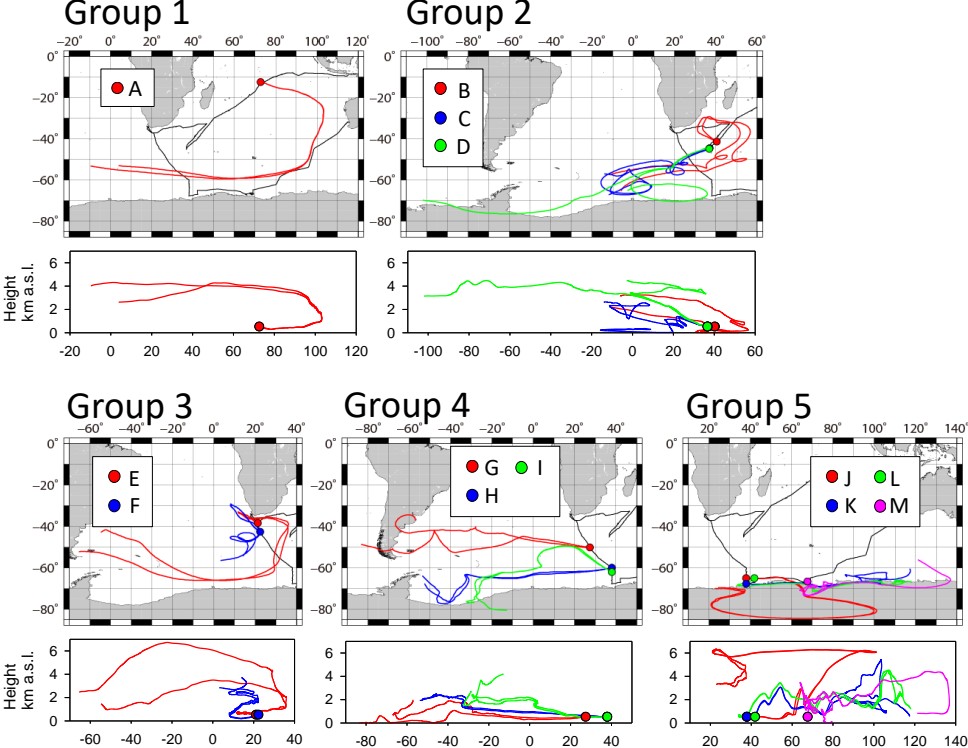

5   **Figure 8: TEM sampling location A–M and the 10-day backward trajectories for air masses reaching the location at 500 m a.s.l.
during sampling periods. Black lines show the ship track. Colored (red, blue, green and pink) circles and lines indicate sampling
locations and backward trajectories, respectively.**





**Figure 9:** Electron micrographs for (a) samples A, C, E, H, J and M, and (b) before and after dialysis of sample J for the same sample region. Magnifications of all microphotographs of (a) are the same. Red arrows indicate examples of particles having sea salt shape. Blue arrows indicate examples of particles having sulfate particle shape. Particles with a satellite structure in samples H and J are marked by green arrows. See main text.



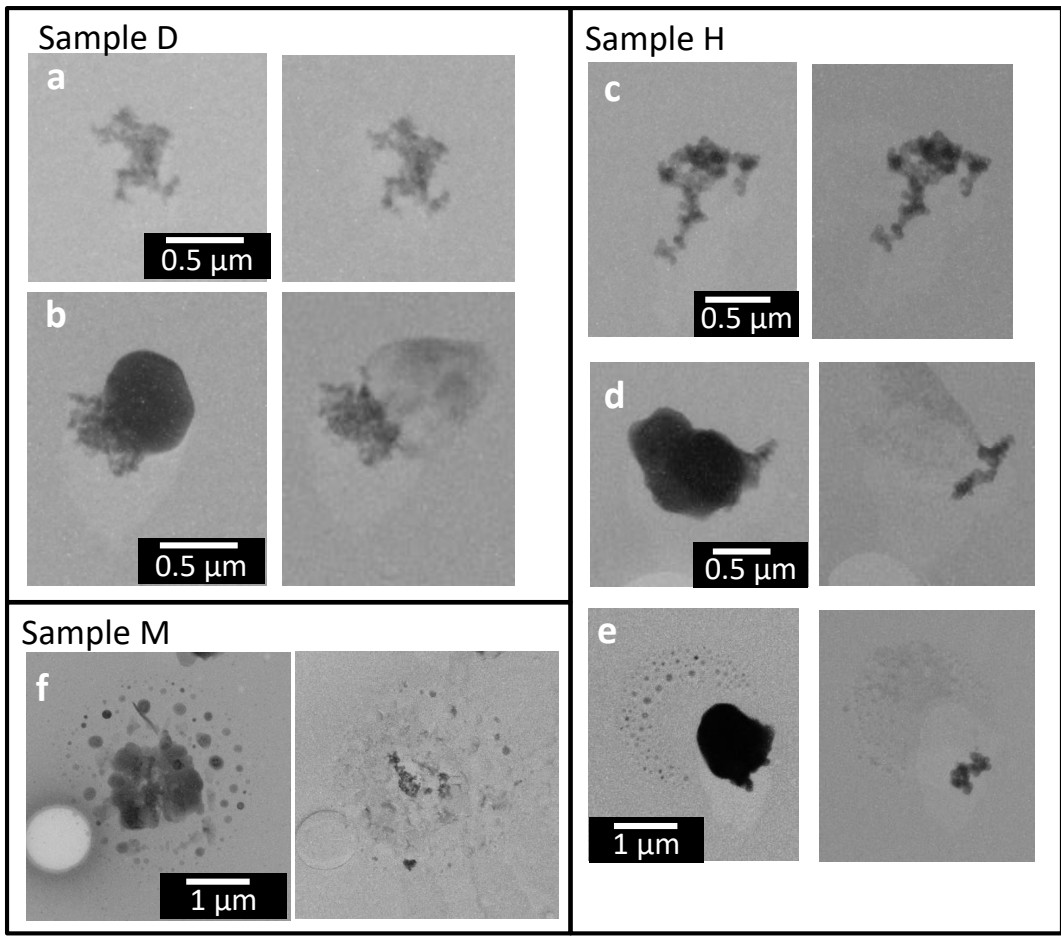

**Figure 10: Electron micrographs of soot-containing particles before (left) and after (right) water dialysis at the same magnification for samples D, H, and M.**



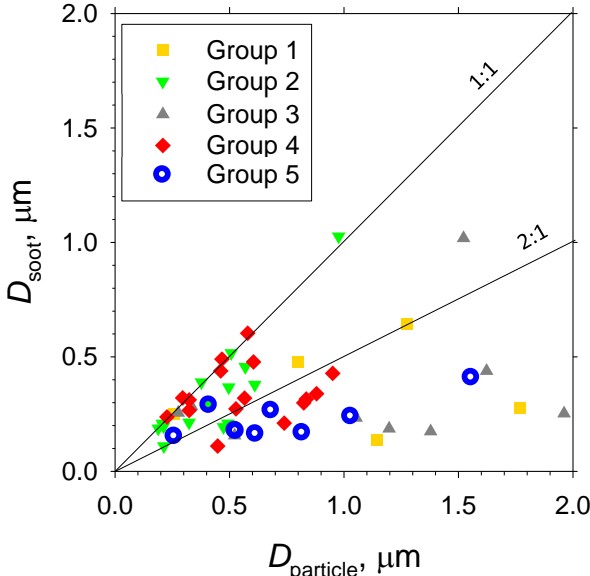

**Figure 11: Scatter plot of particle sizes before and after water dialysis of soot-containing particles (i.e. soot-containing particle diameter and soot diameter) for sample groups 1–5.**



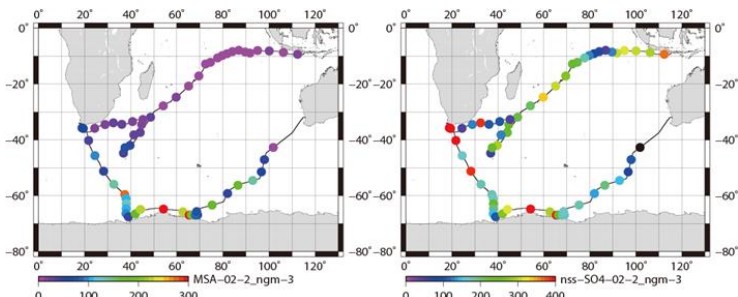

5 **Figure 12: Horizontal variation of mass concentrations of (a) CH$_3$SO$_3^-$ for 0.2–2 µm aerosols and (b) nss-SO$_4^{2-}$ for 0.2–2 µm aerosols.**





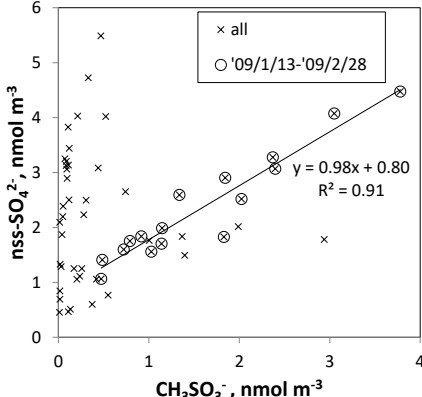

**Figure 13: Scatter plot of CH₃SO₃⁻ and nss-SO₄²⁻ for 0.2–2 µm aerosols. Cross dots and circle dots respectively represent data for all samples and for samples collected at site of larger than 60°S (from 09'1/13 to 09'2/28). The fitting line is linearized for circle dots.**



**Table 1: TEM samples used for this study and analyzed particle and soot-containing particle numbers**

| sample ID | A | B | C | D | E | F | G | H | I | J | K | L | M |
|---|---|---|---|---|---|---|---|---|---|---|---|---|---|
| date | 2008 12/13 | 2008 12/20 | 2008 12/21 | 2008 12/22 | 2009 1/8 | 2009 1/9 | 2009 1/10 | 2009 1/13 | 2009 1/14 | 2009 1/16 | 2009 1/17 | 2009 1/19 | 2009 1/26 |
| start time | 9:13 | 14:52 | 12:14 | 5:25 | 14:45 | 6:29 | 16:33 | 9:56 | 9:56 | 0:50 | 12:35 | 10:37 | 5:49 |
| stop time | 9:43 | 15:27 | 12:34 | 5:55 | 15:25 | 6:49 | 16:53 | 10:31 | 10:51 | 2:00 | 13:40 | 11:37 | 6:49 |
| period (min) | 30 | 35 | 20 | 30 | 40 | 20 | 20 | 35 | 55 | 70 | 65 | 60 | 60 |
| latitude (°S) | 12.5 | 41.3 | 44.6 | 44.9 | 38.5 | 42.6 | 50.2 | 60.0 | 62.0 | 65.0 | 67.7 | 65.0 | 66.6 |
| longitude (°E) | 72.7 | 40.4 | 36.9 | 36.8 | 21.2 | 22.4 | 27.5 | 38.0 | 38.1 | 38.0 | 38.0 | 42.0 | 67.7 |
| wind speed (m/s) | 8.6 | 11.5 | 14.3 | 5.8 | 9.1 | 5.8 | 13.4 | 15.9 | 10.3 | 11.5 | 2.3 | 3 | 7.2 |
| **analyzed particle number** | | | | | | | | | | | | | |
| Stage 1 | 271 | 32 | 84 | 139 | 216 | 201 | 412 | 122 | 272 | 96 | 73 | 18 | 78 |
| Stage 2 | 192 | 2116 | 280 | 488 | 963 | 447 | 1478 | 357 | 703 | 706 | 722 | 6252 | 2634 |
| total | 463 | 2148 | 364 | 627 | 1179 | 648 | 1890 | 479 | 975 | 802 | 795 | 6270 | 2712 |
| **analyzed particle diameter (μm)*** | | | | | | | | | | | | | |
| mean | 0.65 | 0.29 | 0.34 | 0.28 | 0.30 | 0.35 | 0.33 | 0.34 | 0.33 | 0.27 | 0.26 | 0.23 | 0.23 |
| 25 percentile | 0.41 | 0.24 | 0.24 | 0.22 | 0.25 | 0.26 | 0.24 | 0.24 | 0.25 | 0.21 | 0.21 | 0.18 | 0.18 |
| 75 percentile | 0.87 | 0.35 | 0.47 | 0.44 | 0.41 | 0.64 | 0.48 | 0.52 | 0.49 | 0.38 | 0.37 | 0.30 | 0.31 |
| **soot-containing particle number*** | | | | | | | | | | | | | |
| total | 5 | 7 | 2 | 5 | 8 | 2 | 3 | 10 | 4 | 3 | 1 | 2 | 3 |
| bare type | 1 | 1 | 1 | 2 | 2 | 0 | 1 | 4 | 1 | 0 | 0 | 0 | 0 |
| mixed type | 4 | 6 | 1 | 3 | 7 | 2 | 2 | 6 | 3 | 3 | 1 | 2 | 3 |
| satellite type | 0 | 0 | 0 | 0 | 0 | 0 | 0 | 1 | 2 | 1 | 0 | 1 | 3 |
| number fraction of soot-containing-particle to all particle (%)* | 1.1 | 0.33 | 0.55 | 0.80 | 0.68 | 0.31 | 0.16 | 2.1 | 0.41 | 0.37 | 0.13 | 0.03 | 0.11 |

5 *total data for stages 1 and 2



**Table 2: Parameters of the soot-containing particles of TEM samples**

| sample | Group 1 A | | Group 2 B, C, and D | | Group 3 E and F | | Group 4 G, H, and I | | Group 5 J, K, L, and M | |
|---|---|---|---|---|---|---|---|---|---|---|
| Number and percentage of each type soot-containing particles | | | | | | | | | | |
| | number | % | number | % | number | % | number | % | number | % |
| total | 5 | | 14 | | 10 | | 16 | | 9 | |
| bare type | 1 | 20 % | 4 | 29% | 2 | 20% | 6 | 38% | 0 | 0% |
| mixed type | 4 | 80% | 10 | 71% | 8 | 80% | 10 | 62% | 9 | 100% |
| satellite type | 0 | 0 % | 0 | 0 % | 0 | 0 % | 3 | 19 % | 5 | 56 % |
| Averaged values of soot-containing particle diameter ($D_{particle}$) and soot diameter ($D_{soot}$) | | | | | | | | | | |
| $D_{particle}$ (µm) | 1.05±0.50 | | 0.43±0.20 | | 1.10±0.57 | | 0.57±0.22 | | 1.86±2.36 | |
| $D_{soot}$ (µm) | 0.36±0.18 | | 0.34±0.22 | | 0.33±0.25 | | 0.34±0.12 | | 0.23±0.08 | |
| $D_{soot}/D_{particle}$ | 0.47±0.31 | | 0.78±0.22 | | 0.41±0.30 | | 0.68±0.30 | | 0.31±0.22 | |

± standard deviation