# Peer review of "Morphological features and mixing states of soot-containing particles in the marine boundary layer over the Indian and Southern Oceans"

_Atmospheric Chemistry and Physics, 2017_

## Referee Comment (RC1) · Anonymous Referee #1 · 17 Feb 2018

General comments:

The manuscript by Ueda et al, "Morphological features and mixing states of soot-containing particles in the marine boundary layer over the Indian and Southern Oceans" reports aerosol measurements especially soot-containing particles collected from a research cruise. They used various measurement techniques including PSAP, CN counter, OPC, Ion chromatography, and transmission electron microscopy. The data set is valuable for global aerosol researches especially for those who study aerosol in remote area. I found this study is valuable and is clearly written with enough data. The microscopic analysis is also important and useful to understand the mix-

ing states and hygroscopicity of aerosol particles. I have some technical comments to clarify the manuscript.

Page4 Line21-26: It is not clear how the volumes and projected diameter were determined from the projected area S. Please explain the method more detail including how Pt/Pd shadowing was used in this study.

Page7 Line4 "Shapes of particles in these samples were round, dome-like, or rotundate rectangular (examples indicated by blue arrows in Fig. 9).": Blue arrows should indicate sulfate based on the caption in Fig. 9. It is not clear which arrows we should see (and which particles).

Page7 Line13-14 "in the samples H–J collected over the Southern Ocean, as indicated by green arrows": I can not see the satellite structures in the particles indicated by green arrows in Fig 9.

Page7 Line18 "such particles would be acidic droplet particles neutralized by the addition of ammonium after collection": This discussion is not clear. Do you mean the particle changes its shape after collection because of neutralization over the substrate? Some additional explanation or discussion is needed here.

Page7 Line32: "externally mixing of soot" will be "external mixture of soot"

Page23 Fig.9 sample H: The red arrow indicates no particle. Please revise the figure.

---

## Referee Comment (RC2) · Anonymous Referee #2 · 28 Feb 2018

This study provides a careful analysis of individual aerosol particles collected over remote areas of the Indian and Southern Oceans during a long cruise, sections of which could be classified according to the relative amount of anthropogenic influence. The focus is on the concentration and speciation (internal or external mixing) of soot particles, with TEM analysis of intact and water-dialysed samples supplying the core information. (Water dialysis gives an extra angle of individual-particle properties.) Although the identification of individual particle types is based on morphology and water solubility (with no supporting direct compositional information, such as EDS), given the relatively simple composition of remote marine aerosol I believe that the identification of soot, sulfate (with variable degree of acidity) and sea salt is reliable. In all, the re-

sults are interesting and useful for understanding remote marine aerosols; however, the paper is rather descriptive and leaves the reader in doubt about the significance of the results. My questions and comments below address the points that I think the authors may wish to consider, and perhaps to add some more value to their work.

1) The images of sample regions before and after water dialysis are spectacular - however, the quality of the presented TEM images does not seem to permit the detection of very small (consisting of just a few globules, smaller than about 100 nm) soot particles, begging the question whether a fraction of soot particles could escape attention. Could you please comment on the lower size limit of soot that you think you could still identify? On the other hand, if you are confident that only 1 to 2% soot-bearing particles occur, could you please comment on the possible causes of the difference between your and earlier results that showed a larger fraction of soot-bearing sulfates (for example, Pósfai et al. 1999, cited elsewhere, identified in pristine Southern Ocean air 10 to 45% of sulfate internally mixed with soot).

2) Data screening - periods with contamination from the ship were identified by sudden increases in CN counts. Can you absolutely exclude the possibility of enhanced particle counts from natural sources, such as new particle formation followed by rapid growth? An example: comparing Figs. 4a and 4b, even though the particle volume increased on 08/12/22, absorption remained almost constant.

3) Origin of bare or hardly coated soot seems puzzling (discussion on page 8) - have you considered an upper tropospheric source from aircraft emissions?

4) Towards the end of the Results section I miss some discussion on the significance of your observations - do they change our current understanding of remote marine aerosols and their optical properties? What is the main added information?

Minor issues:

Abstract, lines 19-20: change to "particles were rarely found (2.1%..) containing insoluble residuals.."

The Abstract lists only observations; some interpretation, a statement about the significance of the results is missing from the end.

Introduction, first sentence: it sounds as if atmospheric aerosol were a byproduct of combustion - please reword.

"Information related to mixing states of soot has not been shown" - rather "scarcely shown" - see comment 1) above.

3.3.1 Samples, first line: "13 samples were analyzed using TEM"

3.3.2 Morphological features and mixing states, line 22: " most aerosol particles... were almost water soluble". Unclear what "almost" refers to - almost completely dissolved in water or almost all particles were water-soluble? same section, line 27: "However, that in sample H was 2%." Please correct grammar of sentence.

3.3.3 Features of soot-containing particles, line 32: either "externally mixed" or "external mixing"

3rd line from bottom of page 8: "area larger than 60°S" - meaning unclear

Conclusion, line 10: "The origin of bare soot remains unknown."

---

## Author Comment (AC1) · 22 May 2018

Author's response to reviewer #1

General comment: The manuscript by Ueda et al, "Morphological features and mixing states of sootcontaining particles in the marine boundary layer over the Indian and Southern Oceans" reports aerosol measurements especially soot-containing particles collected from a research cruise. They used various measurement techniques including PSAP, CN counter, OPC, Ion chromatography, and transmission electron microscopy. The data set is valuable for global aerosol researches especially for those who study aerosol in remote area. I found this study is valuable and is clearly written

with enough data. The microscopic analysis is also important and useful to understand the mixing states and hygroscopicity of aerosol particles. I have some technical comments to clarify the manuscript.

Response: We thank anonymous Referee #1 for the many constructive comments related to the overall clarity of the article. We revised the manuscript according to the reviewer's comments. Modified words and sentences are highlighted as red in the text of the revised manuscript.

Technical comment 1: Page4 Line21-26: It is not clear how the volumes and projected diameter were determined from the projected area S. Please explain the method more detail including how Pt/Pd shadowing was used in this study.

Response: Incorrect sentences were accidentally retained here. The Pt/Pd shadowing method was not used for this study. Therefore, these sentences were revised as "The diameter of the equivalent circle was estimated from S" (P4L27).

Technical comment 2: Page7 Line4 "Shapes of particles in these samples were round, dome-like, or rotundate rectangular (examples indicated by blue arrows in Fig. 9).": Blue arrows should indicate sulfate based on the caption in Fig. 9. It is not clear which arrows we should see (and which particles).

Response: The sentence was revised to "Based on the image contrast and shadow of Pt/Pd of the particle, most of the particles were classified as round (r), dome-like (d), or rotundate rectangular (rr) on the film (examples indicated by blue arrows in Fig. 9a)." (P7L6). For blue arrows in Fig. 9, round, dome-like, and rotundate rectangular particles were designated respectively as r, d, and rr. Close-up pictures of some particles were added to Fig. 9. In addition, the location of an arrow pointing a specific particle was rearranged to indicate the particle clearly. An explanation of blue arrows was added to the caption in Fig. 9.

Technical comment 3: Page7 Line13-14 "in the samples H–J collected over the Southern Ocean, as indicated by green arrows": I can not see the satellite structures in the particles indicated by green arrows in Fig 9.

Response: Close-up pictures of particles having satellite structures were added to Fig. 9 of the revised manuscript.

Technical comment 4: Page7 Line18 "such particles would be acidic droplet particles neutralized by the addition of ammonium after collection": This discussion is not clear. Do you mean the particle changes its shape after collection because of neutralization over the substrate? Some additional explanation or discussion is needed here.

Response: We revised the sentence to the following in the revised manuscript: "In this study, some rectangular particles showed a satellite structure, which suggests impact by sulfuric acid droplets. Rectangular particles are usually regarded as fully neutralized ammonium sulfate. Therefore, the existence of such particles invites curiosity. One possibility for the origin is the transformation of acidic particles after collection by neutralization with ambient ammonia over the substrate." (P7L21–24).

Technical comment 5: Page7 Line32: "externally mixing of soot" will be "external mixture of soot"

Response: We revised the text as you suggested (P8L10).

Technical comment 6: Page 23 Fig. 9 sample H: The red arrow indicates no particle. Please revise the figure.

Response: The arrow position was revised.

Please also note the supplement to this comment:
https://www.atmos-chem-phys-discuss.net/acp-2017-1179/acp-2017-1179-AC1-supplement.pdf

---

## Author Comment (AC2) · 22 May 2018

Author's response to reviewer #2

General comment: This study provides a careful analysis of individual aerosol particles collected over remote areas of the Indian and Southern Oceans during a long cruise, sections of which could be classified according to the relative amount of anthropogenic influence. The focus is on the concentration and speciation (internal or external mixing) of soot particles, with TEM analysis of intact and water-dialysed samples supplying the core information. (Water dialysis gives an extra angle of individual-particle properties.) Although the identification of individual particle types is based on morphology and water solubility (with no supporting direct compositional information, such as EDS), given the relatively simple composition of remote marine aerosol I believe that the identification of soot, sulfate (with variable degree of acidity) and sea salt is reliable. In all, the results are interesting and useful for understanding remote marine aerosols; however, the paper is rather descriptive and leaves the reader in doubt about the significance of the results. My questions and comments below address the points that I think the authors may wish to consider, and perhaps to add some more value to their work.

Response: We thank anonymous Referee #2 for the many constructive comments which have improved our manuscript. Modified words and sentences are highlighted as red in the text of the revised manuscript.

Comment 1 The images of sample regions before and after water dialysis are spectacular - however, the quality of the presented TEM images does not seem to permit the detection of very small (consisting of just a few globules, smaller than about 100 nm) soot particles, begging the question whether a fraction of soot particles could escape attention. Could you please comment on the lower size limit of soot that you think you could still identify? On the other hand, if you are confident that only 1 to 2% soot-bearing particles occur, could you please comment on the possible causes of the difference between your and earlier results that showed a larger fraction of soot-bearing sulfates (for example, Pósfai et al. 1999, cited elsewhere, identified in pristine Southern Ocean air 10 to 45% of sulfate internally mixed with soot).

Response: We added "Using this method, insoluble materials of less than 0.1 $\mu$m diameter were not identifiable as soot because of TEM image quality." to the manuscript (P7L31). However, as shown in Fig. 9b, the water dialysis analysis reveals clearly that most particles were composed only of water-soluble materials. Although the value might be underestimated for < 0.1 $\mu$m diameter, we are confident about the estimated fraction of soot-containing particles without burial in a particle after water dialysis. Several observations in moderately remote atmosphere have also revealed the number fraction of soot-containing particles using the same water dialysis analysis (Hasegawa

and Ohta, 2002, Ueda et al. 2011b). The quantities of soot-containing particles in this study (less than 2%) were smaller than their values (3–11% for particles 0.08–1.6 $\mu$m at Fukue Island in northwestern Japan by Hasegawa and Ohta (2002) and 2–25% for particles 0.2–0.4 $\mu$m and 14–59% for particles 0.4–0.7 $\mu$m at Cape Hedo in south-western Japan by Ueda et al. (2011b)). For the soot-containing fraction in the remote marine troposphere above the Southern Ocean, Pósfai et al. (1999) reported that 10–45% for particles >0.1 $\mu$m were sulfate particles contained soot inclusions based on TEM analysis. They also identified aircraft emissions and biomass burning as the most likely major sources of soot. However, our study observed aerosols within the marine boundary layer by ship. Most of the sampling sites were distant from continental source areas comparing with previous studies. In addition, the backward air mass trajectories had not passed over continental areas (except Antarctica) for a week. The low fraction of soot-containing particles in this study would be a result of remoteness of the atmosphere observed. We added this discussion to P7L37–P8L5 of the revised manuscript.

Comment 2 Data screening - periods with contamination from the ship were identified by sudden increases in CN counts. Can you absolutely exclude the possibility of enhanced particle counts from natural sources, such as new particle formation followed by rapid growth? An example: comparing Figs. 4a and 4b, even though the particle volume increased on 08/12/22, absorption remained almost constant.

Response: The first author has studied new particle formation over the Pacific Ocean, using a similar screening method (Ueda et al., 2016a). Screening threshold of increase rate of particle number concentration (1.2 times per 10 min) is sufficiently higher than that of naturally observed increasing rate for typical new particles formation over the ocean (Ueda et al., 2016a). Therefore, data on new particle formation would remain after our data screening. We added this explanation to P3L24–28. As exemplified on 08/12/22, wind speed and the number concentration in larger size range were high for the day, suggesting high sea salt loading in the lower atmosphere. This period was not

excluded by our data screening because increase rate of particle number concentration was below 1.2 times per 10 min.

Comment 3 Origin of bare or hardly coated soot seems puzzling (discussion on page 8) – have you considered an upper tropospheric source from aircraft emissions?

Response: Most backward trajectories for samples of groups 1–4 were passed below 2 km a.s.l. during 5 days before sample collection. In addition, most of the horizontal backward trajectories did not pass civil aviation major route of Stettler et al. (2013) during the several days preceding the sampling. However, we cannot exclude the slight possibility of the soot contribution from aircraft. We added a discussion related to the possibility of an upper tropospheric source from aircraft emissions (P8L31–42).

Comment 4 Towards the end of the Results section I miss some discussion on the significance of your observations - do they change our current understanding of remote marine aerosols and their optical properties? What is the main added information?

Response: This study specifically addressed individual features of soot-containing particles at a remote marine boundary layer, which was distant from the source regions of soot. Main new information is mixing states of soot-containing particles over the Antarctic Ocean. Although bare soot particles were found over the Indian Ocean and northern parts of the Southern Ocean, all soot containing particles collected near the Antarctic coast were mixed with water-soluble materials. We revised some parts of the Results (P7L37–P8L5, P9L13, and P9L31) and Conclusions (P9L41– P10L14), to clarify our findings and their significance.

Minor issues: Comment: Abstract, lines 19-20: change to "particles were rarely found (2.1%..) containing insoluble residuals.."

Response: The sentence was revised according to your comment.

Comment: The Abstract lists only observations; some interpretation, a statement about the significance of the results is missing from the end.

Response: The last sentence of the Abstract was revised. Additionally, we made minor revisions to the Abstract to adjust the word number.

Comment: Introduction, first sentence: it sounds as if atmospheric aerosol were a byproduct of combustion - please reword.

Response: We modified the text as "Soot in the atmospheric aerosol is a by-product of fossil fuel (diesel and coal) combustion and open biomass burning. It is a carbonaceous material. . .."

Comments: "Information related to mixing states of soot has not been shown" - rather "scarcely shown" - see comment 1) above.

3.3.1 Samples, first line: "13 samples were analyzed using TEM"

Response: These sentences were changed according to your comments (P2L17 and P6L33).

Comment: 3.3.2 Morphological features and mixing states, line 22: " most aerosol particles. . . were almost water soluble". Unclear what "almost" refers to - almost completely dissolved in water or almost all particles were water-soluble?

Response: The word "almost" was removed (P7L28).

Comment: same section, line 27: "However, that in sample H was 2%." Please correct grammar of sentence.

Response: The sentence with relating part was revised as below. "The number fractions of soot-containing particles to total particles for all samples except for sample H were 1% or less. The fraction for sample H was 2%." (P7L33–34)

Comment: 3.3.3 Features of soot-containing particles, line 32: either "externally mixed" or "external mixing"

Response: This sentence was changed to "external mixture of soot", according to the

comment from the reviewer#1 (P8L10).

Comment: 3rd line from bottom of page 8: "area larger than 60âŮęS" - meaning unclear

Response: This sentence was revised as "latitude higher than 60°S" (P9L24).

Comment: Conclusion, line 10: "The origin of bare soot remains unknown."

Response: This sentence was revised as you suggested (P10L1).

Please also note the supplement to this comment:
https://www.atmos-chem-phys-discuss.net/acp-2017-1179/acp-2017-1179-AC2-supplement.pdf

———————————————

---

## Author Response (AR2)

We thank the editor for technical corrections of the article. We revised the manuscript according to your comments. Modified words are highlighted as blue in the text of the revised manuscript.